# Two-dimensional electronic spectroscopy of bacteriochlorophyll *a* with synchronized dual mode-locked lasers

JunWoo Kim [1,4], Jonggu Jeon[1], Tai Hyun Yoon [1,2✉] & Minhaeng Cho [1,3✉]

How atoms and electrons in a molecule move during a chemical reaction and how rapidly energy is transferred to or from the surroundings can be studied with flashes of laser light. However, despite prolonged efforts to develop various coherent spectroscopic techniques, the lack of an all-encompassing method capable of both femtosecond time resolution and nanosecond relaxation measurement has hampered various applications of studying correlated electron dynamics and vibrational coherences in functional materials and biological systems. Here, we demonstrate that two broadband (>300 nm) synchronized mode-locked lasers enable two-dimensional electronic spectroscopy (2DES) study of chromophores such as bacteriochlorophyll *a* in condensed phases to measure both high-resolution coherent vibrational spectrum and nanosecond electronic relaxation. We thus anticipate that the dual mode-locked laser-based 2DES developed and demonstrated here would be of use for unveiling the correlation between the quantum coherence and exciton dynamics in light-harvesting protein complexes and semiconducting materials.

[1] Center for Molecular Spectroscopy and Dynamics, Institute for Basic Science (IBS), Seoul 02841, Republic of Korea. [2] Department of Physics, Korea University, Seoul 02841, Republic of Korea. [3] Deparment of Chemistry, Korea University, Seoul 02841, Republic of Korea. [4]Present address: Department of Chemistry, Princeton University, Princeton, NJ 08544, USA. ✉email: thyoon@korea.ac.kr; mcho@korea.ac.kr

Our understanding of natural light-harvesting systems involving excitonic energy transfers has been revolutionized by the application of femtosecond nonlinear optical spectroscopy[1,2]. Delocalized quantum excited states created by photoexcitation of and electronic couplings between light-absorbing chromophores relax through a myriad of pathways, such as exciton migration, excitation localization induced by the thermal fluctuation of coupled bath degrees of freedom, and long-distance Förster energy transfer[3–5]. A series of events involving electron and proton transfer reactions subsequently occur in photosynthetic systems. One of the most crucial techniques for studying such complex photoinduced reaction dynamics of generic self-assembled multichromophoric systems is two-dimensional electronic spectroscopy (2DES)[1,6–8], which is capable of providing the time correlation between the initial and final states by mapping their nonlinear optical response onto 2D excitation- and detection-frequency space[9,10].

The 2DES requires multiple femtosecond optical pulses to interrogate molecular systems, such as light-harvesting protein complexes[1,11–14], optical chromophores in solutions[10,15], semiconducting materials[16,17], metallic nanoparticles[18], and chiral aggregates[19]. It provides information on not only ultrafast energy transfer kinetics but also dephasing time scales of electronic and/or vibrational coherences of coupled chromophore systems like photosynthetic proteins[12,20]. Recently, the 2DES of Fenna–Matthew–Olson complex revealed an important role of vibronic couplings in the ultrafast energy transfer process on the one-exciton state manifold[6,8,13]. It turns out that the vibronically excited vibrational modes play a crucial role in enhancing the pigment-to-pigment energy transfer rates[21,22]. In addition, the excited-state dynamics and relaxations of chlorophyll molecules that are critical pigments in photosynthetic complexes are of fundamental importance[23–25].

In 2DES experiments, the time evolution of excited states initiated by a pair of pump pulses generating temporally interfered electric fields is monitored by using another time-delayed probe and local oscillator pulses. The waiting time ($T_w$) that is the time delay between the pump and probe pulses is usually scanned by mechanically changing the optical path lengths of the excitation and detection pulses. It was shown that the $T_w$-scanning speed can be dramatically enhanced by utilizing two repetition rate-stabilized mode-locked lasers (MLs), where the technique is called asynchronous optical sampling (ASOPS)[26]. In ASOPS, each ML generates a train of optical pulses separated by an equal time interval. When the repetition rates of the two MLs are slightly detuned as $f_r$ and $f_r + \Delta f_r$, the optical time delay ($T_w$) between a pair of pulses from the two MLs increases linearly in laboratory time ($t$). The corresponding time delay increment ($\Delta T_w$) is determined by the repetition rate and the detuning factor $\Delta f_r$ as

$$\Delta T_w = \frac{1}{f_r} - \frac{1}{f_r + \Delta f_r} \cong \frac{\Delta f_r}{f_r^2} = f_D \frac{1}{f_r}. \quad (1)$$

Here, $f_D$ is the down-conversion factor defined by $\Delta f_r / f_r$. Although the automatic time-delay scanning scheme referred to as ASOPS was proposed and demonstrated in 1987 (ref. [26]), the application of ASOPS to coherent multidimensional spectroscopy has not been achieved, even though the ASOPS has been employed in the development of dual-frequency comb spectroscopic techniques[27,28]. Recently, synchronized MLs without photodetector array nor mechanical time-delay devices have been used to perform Raman microspectroscopy[29,30], dual-frequency comb 2DES[31], and time- and frequency-resolved transient absorption and refraction spectroscopy[32–34].

Here, we experimentally demonstrate that synchronized mode-locked laser-based 2DES (SM-2DES) without a long mechanical delay line for the $T_w$−scan enables ultrafast time-resolved studies of sub-picosecond electronic and vibrational dephasings,

picosecond solvation dynamics, and nanosecond population relaxation without sacrificing its femtosecond time-resolvability and detection sensitivity. We show that the data acquisition time for the measurement of the $T_w$-dependent 2DES signals is substantially shortened in comparison to the conventional 2DES technique. We, here, present the SM-2DES experimental results for bacteriochlorophyll $a$ (BChla)/1-propanol solution to demonstrate the feasibility and capability of a novel coherent 2D spectroscopic technique. It should be noted that the ASOPS-based femtosecond time-resolved spectroscopy with an interferometric detection of the nonlinear signal[34], including SM-2DES, requires tightly repetition-rate-stabilized MLs to achieve a sub-fs timing jitter. The fast and efficient data acquisition rate of SM-2DES enabled us to study ultrafast solvation dynamics, vibrational coherences of vibronically coupled modes, and non-Condon effects on the coherent vibrational spectra extracted from the SM-2DES signals.

## Results

**Synchronized mode-locked laser-based 2DES**. The experimental setup for SM-2DES is illustrated in Fig. 1a (see Supplementary Note 1 for detailed optical layout). SM-2DES in a boxcar geometry requires four temporally and spatially separated pulses that are referred to as their wave vectors $\mathbf{k}_A$, $\mathbf{k}_B$, $\mathbf{k}_C$, and $\mathbf{k}_{LO}$, where LO represents the local oscillator used to interferometrically detect the generated four-wave-mixing signal electric field[10]. The repetition rates of $ML_1$ and $ML_2$ are slightly different from each other by $\Delta f_r$ (Fig. 1a), where the repetition rate ($f_r$) of $ML_2$ is 80 MHz, and that of $ML_1$ is 80 MHz + $\Delta f_r$. $\Delta f_r$ was chosen as 38.4 Hz and 3.2 kHz to set $\Delta T_w = 6$ fs and 500 fs, respectively, according to Eq. (1). First, two pump pulses with wave vectors $\mathbf{k}_A$ and $\mathbf{k}_B$, respectively, that are used to excite chromophores in solutions are produced by $ML_1$, and the time delay between the two pulses, denoted as $\tau_1$, is experimentally controlled using a high-precision translational stage. This spatial interference fringe generated by the two pump fields propagating in the two different directions determined by the wave vectors of $\mathbf{k}_A$ and $\mathbf{k}_B$ acts like a grating. Then, the probe pulse propagating along the direction of $\mathbf{k}_C$ arrives at the sample after a waiting time $T_w$, and then the probe beam is scattered due to the third-order field–matter interaction. Such a three-pulse scattering process was initially studied by Weiner and Ippen[35]. The $T_w$-dependent relaxation of the transient grating diffracted in the direction of $\mathbf{k}_{sig}$ ($= -\mathbf{k}_A + \mathbf{k}_B + \mathbf{k}_C$) results from the decay of the population of excited-state molecules in a dissipative medium. The scattered signal is under interferometric detection with an added LO pulse (with wave vector $\mathbf{k}_{LO} = \mathbf{k}_{sig}$) from $ML_2$. Here, the delay time between the probe and LO pulses, denoted as $\tau_2$, is scanned using another high-precision translational stage.

Figure 1b depicts one of the double-sided Feynman diagrams[36,37] describing a photon-echo-process contributing to the 2DES signal, where $|g\rangle$ is the electronic ground state and $|e\rangle$ and $|f\rangle$ denote two different vibrational states in the electronically excited state. Because the signal field is interferometrically detected with LO field from $ML_2$, the measured 2DES signal becomes independent of carrier-envelope phase slips of the two MLs[38]. As can be seen in the energy level diagram and comb structures of the pulse spectra in Fig. 1c and d, respectively, the vibrational coherence with a frequency of $\nu_{gf} - \nu_{fe} \approx (-n + m)f_r$ can be measured by allowing the third-order 2DES signal field to interfere with a specific LO comb mode, which results in a beat signal with an RF of $(-n + m)\Delta f_r$. Here, $n$, $m$, and $p$ are the frequency comb mode numbers.

Figure 2a shows a representative set of SM-2DES raw data of the IR125/ethanol solution. When the pump pulse with wave vector $\mathbf{k}_A$ precedes the $\mathbf{k}_B$ pulse, the photon echo response dominates the detected 2DES signal for an optically

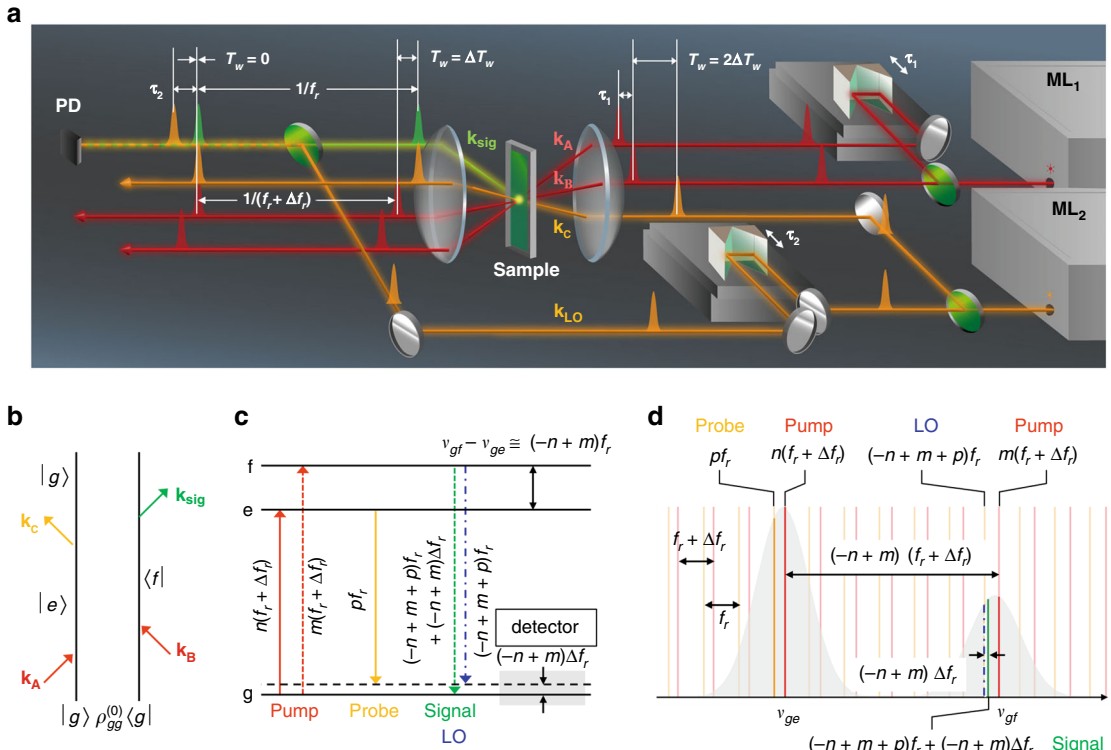

**Fig. 1 Synchronized mode-locked laser-based two-dimensional electronic spectroscopy. a** Schematic representation of the SM-2DES experimental setup. Green circular plates represent beam splitters. The $ML_1$ beam is split into two beams with wave vectors $\mathbf{k}_A$ and $\mathbf{k}_B$, and the time delay $\tau_1$ between the two pulsed beams is scanned using a translational stage. The pulses propagating along the directions determined by $\mathbf{k}_A$ and $\mathbf{k}_B$ are used as a pair of pump pulses to excite optical chromophores in solution. The $ML_2$ beam is split into two beams with wave vectors $\mathbf{k}_C$ and $\mathbf{k}_{LO}$. The time delay $\tau_2$ between the $\mathbf{k}_C$ and $\mathbf{k}_{LO}$ pulses is scanned using another translational stage. The transient grating generated by $\mathbf{k}_A$ and $\mathbf{k}_B$ beams diffracts the probe with wave vector $\mathbf{k}_C$ into the direction satisfying the desired phase-patching condition $\mathbf{k}_{sig} = -\mathbf{k}_A$ ($ML_1$) $+ \mathbf{k}_B$ ($ML_1$) $+ \mathbf{k}_C$ ($ML_2$), where the 2DES signal under detection propagates along the direction determined by $\mathbf{k}_{sig}$. The repetition frequencies of the pump and probe beams are $f_r + \Delta f_r$ and $f_r$, respectively. The 2DES signal is combined with the local oscillator field with a wave vector $\mathbf{k}_{LO}$ that is collinear with the propagation direction of the 2DES signal electric field, and the interference signal is detected by a photodetector (PD). **b** Double-sided Feynman diagram describing one of the various terms contributing to the 2DES signal. **c** Energy-level diagram of a three-level system. $|g\rangle$ is the ground state, and $|e\rangle$ and $|f\rangle$ represent two vibrational states in the electronically excited state; $n$, $m$, and $p$ are the relevant frequency comb mode numbers. $\nu_{ge}$ ($\nu_{gf}$) indicates the optical transition frequency between $|g\rangle$ and $|e\rangle$ ($|f\rangle$) states. The vibrational quantum beat with a frequency of $\nu_{gf} - \nu_{fe} \approx (-n + m)\,f_r$ is measured through the interferometric detection of the 2DES signal with LO from $ML_2$. The beat signal contributing to the time-domain interferogram oscillates with a frequency of $(-n + m)\Delta f_r$, which is in the RF domain. **d** Comb structures of the pump, probe, and LO fields in the frequency domain. The line-broadened absorption bands of the $|g\rangle - |e\rangle$ and $|g\rangle - |f\rangle$ transitions are shown in grey.

heterogeneous system. The opposite time-ordered field-matter interactions in the order of $\mathbf{k}_B \rightarrow \mathbf{k}_A \rightarrow \mathbf{k}_C$ pulses generate the non-rephasing 2DES signal. At a fixed $\tau_1$, a continuous $\tau_2$-scan is performed while ASOPS with two trains of $\mathbf{k}_B$ and $\mathbf{k}_C$ pulses from the two MLs is used to scan the waiting time ($T_w$) that is related to laboratory time $t$ by $T_w = f_D \Delta t$. $\Delta t$ is the time elapsed after the optical trigger signal generated when a pair of pulses from the two MLs temporally overlap. The most prominent interference pattern of the SM-2DES signal at $\tau_1 = 0$ shown in Fig. 2b arises from the interference between the local oscillator electric field and the 2DES signal electric field. However, the interferogram in Fig. 2b depends on $T_w$ too, even though it is not clearly visible in the interferogram plotted in Fig. 2b. One can retrieve the $T_w$-dependent 2DES signals by rearranging the raw data. The $T_w$-dependent feature of the SM-2DES signal can be clearly seen in the zoomed-in plot in Fig. 2c. Each decay curve (Fig. 2d) at fixed $\tau_1$ and $\tau_2$ corresponds to the measurement (or waiting) time-dependent interference signal, i.e., $I(\tau_2, \Delta t, \tau_1) = I(\tau_2, T_w / f_D, \tau_1)$, which shows how the vibrational quantum beats in the infrared frequency domain can be measured with a photodetector in the RF domain by employing an ASOPS-based down-conversion measurement scheme. The experimental data in Fig. 2a is,

therefore, a 3D time-domain interferometric signal $I(\tau_2, \Delta t, \tau_1)$ that depends on $\tau_1$, $\Delta t$ ($= T_w / f_D$), and $\tau_2$.

To establish the relationship between the measured time-domain interferogram in Fig. 1b and the nonlinear response function $S^{(3)}(t_3, t_2, t_1)$ of optical chromophores in condensed phases, we employ time-dependent perturbation theory[37,39], use optical frequency comb field expressions[38], and invoke an impulsive pulse approximation. One can find that the experimentally measured signal $I(\tau_2, \Delta t, \tau_1)$ that results from the interference between the LO and 2DES signal electric fields, $\mathbf{E}_{LO}$ and $\mathbf{E}_{signal}$, respectively, is given by (see Supplementary Note 2 for detailed theoretical derivation and ref. [40]):

$$I(\tau_2, \Delta t, \tau_1) \propto \mathrm{Re}\left[\mathbf{E}_{LO}^*(t) \times \mathbf{E}_{signal}^*(t)\right] \propto \mathrm{Re}[R_{echo}(\tau_2, T_w, \tau_1)],$$
$$(2)$$

where $R_{echo}(\tau_2, T_w, \tau_1)$ is the time-domain nonlinear response function associated with photon-echo spectroscopy[41]. Equation (2) suggests that the all-time-resolved SM-2DES interferogram is simply proportional to the real part of the photon-echo response function. As manifested in Eq. (2), the time scale of femtosecond and nanosecond molecular processes during $T_w$ is converted to

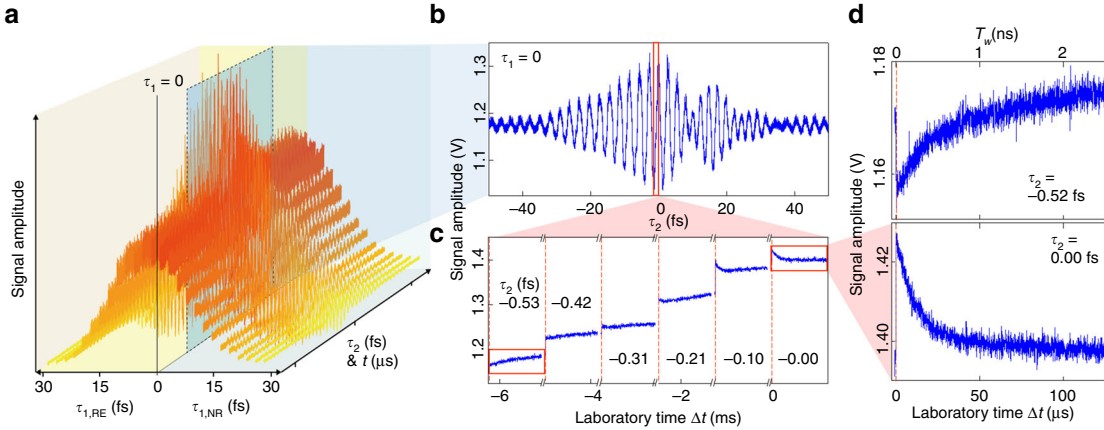

**Fig. 2 Time-domain interferometric 2DES signal. a** Experimentally measured SM-2DES raw data of IR125/ethanol solution with respect to $\tau_1$ and $\tau_2$. Here, the repetition frequency detuning factor $\Delta f_r$ is 1.6 kHz. **b** SM-2DES interference signal with respect to $\tau_2$ when $\tau_1 = 0$ fs. The $T_w$-dependent decaying signals are hidden in the prominent $\tau_2$-dependent interference signal. **c** Zoomed-in signal taken from the red box in **b**. **d** Zoomed-in $T_w$-dependent signals $\tau_2 = -0.52$ fs (top) and $\tau_2 = 0.00$ fs (bottom) at $\tau_1 = 0$ fs.

that of laboratory time $\Delta t$ on the nanosecond to millisecond time scales by the inverse of the down-conversion factor, i.e., $1/f_D = 2.5 \times 10^4$ and $2.08 \times 10^6$ for $\Delta f_r = 3.2$ kHz and 38.4 Hz, respectively, with $f_r = 80$ MHz in the present work.

Fourier-transforming the 3D interferometric signal, $I(\tau_2, \Delta t, \tau_1)$, with respect to $\tau_1$ and $\tau_2$, we were able to obtain the 2DES spectra with respect to the excitation angular frequency ($\omega_1$) and detection angular frequency ($\omega_2$). A notable advantage of SM-2DES compared to the conventional 2DES is that both the amplitude and phase of the 2DES data can be analytically normalized and corrected, respectively, by using the interferometric pump-probe signals measured with the same instrument (see Methods and Supplementary Note 3). Here, the phase stability between the probe and LO pulses is maintained by employing a continuous $\tau_2$-scan method[34] that is combined with an additional reference interferometer (see Methods). In practice, it should be noted that the present phase correction and spectral amplitude calibration methods are useful when the complex (real and imaginary) pump-probe data are available. However, the spectral shapes of the pump pulses are unavoidably imprinted in the measured SM-2DES spectra. Since the automatic $T_w$-scan was performed up to 3.0 ns with a time interval of 6 fs or 500 fs, when the repetition rate detuning factor is 38.4 Hz or 3.2 kHz, respectively, the sheer size and information density of 2DES data (many thousands to millions of 2DES spectra) are extraordinarily large, which attests to the capability of our SM-2DES technique.

**SM-2DES of bacteriochlorophyll _a_.** The spectral bandwidth of our Ti:Sapphire oscillators is broad enough to cover both the $Q_y$-absorption and -emission bands of the BChl$a$ in propanol as well as the high-lying vibronic bands (Fig. 3a). Three representative transitions between the ground ($S_0$) and $Q_y$ states, which are the 0–0 $Q_y$ absorption ($Q_{y,0}$) at 387 THz, 0–1 $Q_y$-absorption ($Q_{y,1}$) at 420 THz, and 0–0 $Q_y$-emission at 375 THz, are marked by dashed lines. The $Q_{y,0}$-emission peak is Stokes-shifted from the $Q_{y,0}$-absorption peak due to free energy minimization processes such as solvation. The time resolution of our SM-2DES is 12.5 fs (see Method and Supplementary Note 1) so that one can study the wave packet oscillations of vibronically coupled modes with frequencies up to 2670 cm$^{-1}$. During the time required for a full $T_w$-scan, which is $1/\Delta f_r$, the number of $T_w$-points in the dynamic range of the $T_w$-resolved measurements from 0 to 12.5 ns is $2.08 \times 10^6$ when $\Delta f_r = 38.4$ Hz and $f_r = 80$ MHz and $2.5 \times 10^4$ when $\Delta f_r = 3.2$ kHz and $f_r = 80$ MHz. However, the conventional

2DES with a mechanical delay line for the $T_w$-scan is, in practice, not useful to collect such a large number of $T_w$-dependent 2DES signals. Due to the high data acquisition rate of the SM-2DES compared to the conventional 2DES methods using a single ML and a mechanical delay line for scanning the waiting time, the frequency resolution of the coherent vibrational spectrum extracted from the time-resolved 2DES signals can be substantially enhanced. Furthermore, the large number of data equally spaced over a wide range of waiting time enables to perform boxcar averaging, which can, in turn, enhance the signal-to-noise ratio of the 2DES spectrum even at a long waiting time at which the amplitude of the signal is expected to be small.

In Fig. 3b, the six representative 2DES spectra at waiting times of 50 fs, 500 fs, 5 ps, 50 ps, 500 ps, and 3.0 ns are shown. At a short waiting time ($T_w = 50$ fs), both ground-state bleach (GSB) and stimulated emission (SE) positively contribute to the absorptive 2DES signal. In general, the coherent electronic artefact significantly contributes to the 2DES signal at zero waiting time when the three pulses that interact with the sample overlap in time[38]. The diagonal peak at (387 THz, 387 THz) and the cross-peak at (420 THz, 387 THz) originating from both the GSB and SE terms become vertically elongated along the detection frequency due to rapid intramolecular vibrational redistribution and solvation dynamics, which causes the detection frequency of the SE contribution to shift down to 375 THz in less than 5 ps[37]. The two GSB contributions denoted as GSB$_0$ at (387 THz, 387 THz) and GSB$_1$ at (420 THz, 387 THz) result from the GSBs produced at $Q_{y,0}$ by the $Q_{y,0}$- and $Q_{y,1}$-excitations, respectively. The $Q_{y,1}$ diagonal peak at (420 THz, 420 THz) is not clearly visible in Fig. 3b because its amplitude is much smaller than the $Q_{y,0}$ diagonal peak, denoted as GSB$_0$, by a factor of ~0.06. Also, there should be another GSB peak at (387 THz, 420 THz), which results from the GSB of $Q_{y,1}$ state when the pump induced a $Q_{y,0}$ transition. The amplitude of the cross-peak at (387 THz, 420 THz) could be comparable to that of GSB$_1$. However, the GSB$_1$ peak is found to be approximately four times stronger than that of the cross-peak at (387 THz, 420 THz) (Supplementary Fig. 2c). Such a reduction in the amplitude of the cross-peak at (387 THz, 420 THz) is caused by the negative contribution from an excited-state absorption[25]. The two SE contributions denoted as SE$_0$ at (387 THz, 375 THz) and SE$_1$ at (420 THz, 375 THz) result from the Stokes-shifted excited states $Q_{y,0}$ and $Q_{y,1}$, respectively. The GSB$_1$ and SE$_1$ in Fig. 3b are frequency-resolved from the GSB$_0$ and SE$_0$ peaks (Fig. 3a), even though both the absorption and emission spectra have broad lineshapes

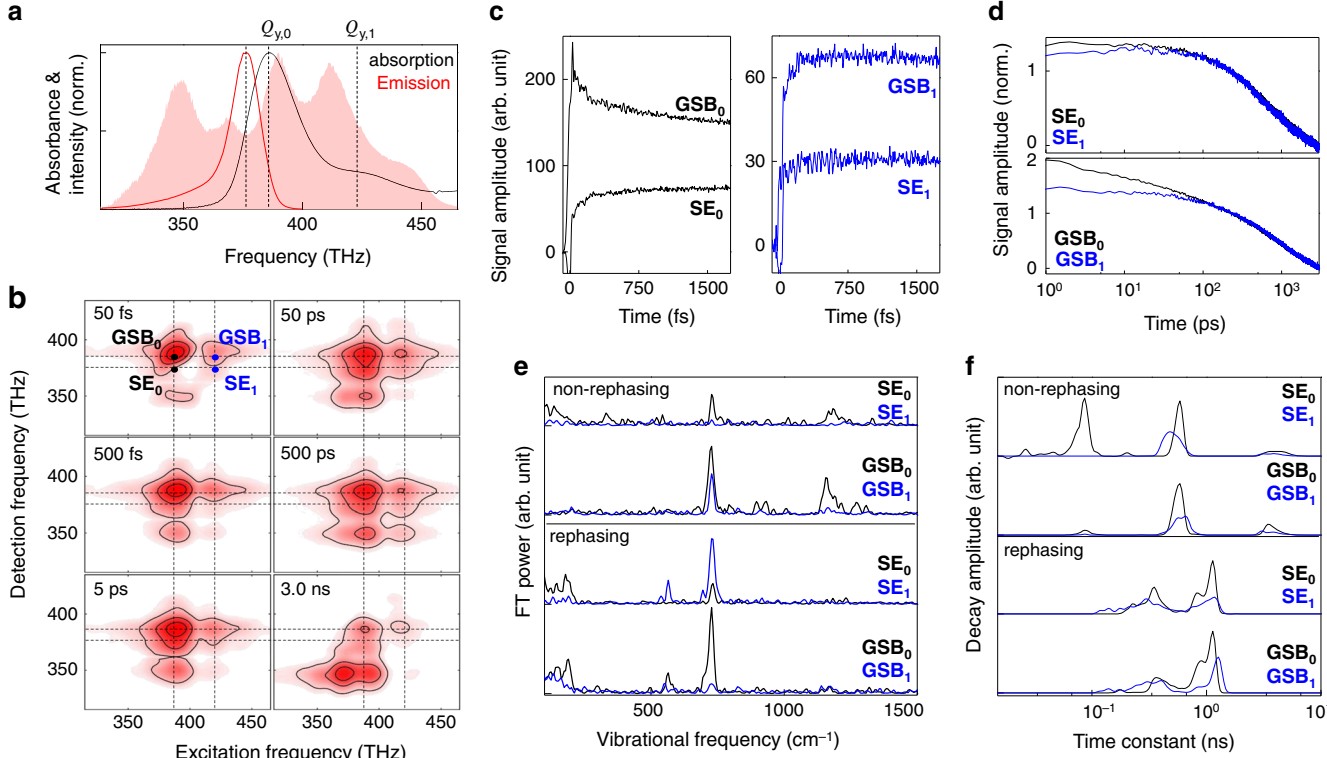

**Fig. 3 Time-resolved SM-2DES spectra of BChla. a** The absorption (black) and emission (red) spectra of BChla/1-propanol solution. The power spectrum of the light source (pink area) is presented for the sake of direct comparison. **b** Absorptive (rephasing + non-rephasing) SM-2DES spectra of the BChla solution at $T_w$ = 50, 500 fs, 5 ps ($\Delta f_r$ = 38.4 Hz), 50 ps, 500 ps and 3 ns ($\Delta f_r$ = 3.2 kHz). Each spectrum is normalized by its maximum amplitude. **c, d** The time traces of the rephasing 2DES signals at four different positions in the SM-2DES spectra, which are **$SE_0$** (387 THz, 375 THz), **$GSB_0$** (387 THz, 387 THz), **$SE_1$** (420 THz, 375 THz), and **$GSB_1$** (420 THz, 387 THz). **e** Fourier transforms of the oscillating components in the time traces plotted in **c**. **f** The distribution of decay time constants associated with the time traces shown in **d**.

with a shoulder peak in the absorption spectrum that originates from a vibronic progression. Such clear separations of the $GSB_1$ and $SE_1$ peaks from the $GSB_0$ and $SE_0$ peaks in the frequency domain are partly due to the non-Gaussian spectral shapes of the pump pulses (Fig. 3a) that affect the lineshape of the 2DES spectrum along the excitation frequency axis. Note that, in general, both the pump and probe spectra affect the spectral features of the 2DES spectrum along the excitation and detection frequencies. Here, we find a positive peak at (387 THz, 350 THz) whose amplitude increases on a time scale of nanoseconds, which could be assigned to the thermal grating contribution that results from nonradiative decay of the excited BChla molecules.

The SM-2DES spectra at long waiting times obtained by employing a fast $T_w$-scan mode ($\Delta f_r$ = 3.2 kHz) (see the 2DES spectra at $T_w$ = 50 ps, 500 ps, and 3.0 ns in Fig. 3b) provide information on slow population relaxation and local heating dynamics. The 2D spectral shapes of the four peaks in the 50-ps 2DES spectrum remain constant up to a waiting time of 500 ps, and the corresponding signals decay to zero on a few nanoseconds time scale. In the 2DES spectrum at $T_w$ = 3.0 ns, a horizontally elongated positive peak appears, which can be assigned to the formation of a transient thermal grating in the solution sample due to the non-collinearly propagating trains of pulses (see Supplementary Note 4). The slow relaxation and evolution of the thermal grating provide useful information on thermal diffusion processes in the surroundings of chromophores in condensed phases.

**Picosecond relaxation and solvation dynamics.** The time profiles of the 2DES signals at the four peak positions ($GSB_0$, $SE_0$, $GSB_1$, and

$SE_1$) over the broad range (six decades) of waiting times are plotted in Fig. 3c and d. The amplitudes of the slow time profiles from 200 ps to 3 ns are matched with each other so that one can directly compare the early parts of the four decaying signals with $T_w$ (Fig. 3d). At short waiting times (<100 ps), the decay profiles of $SE_0$ and $GSB_0$ that are associated with the $Q_{y,0}$ excitation differ from those of the $SE_1$ and $GSB_1$ signals, respectively (Fig. 3c and d). Recently, Moca et al. carried out a series of 2DES experiments for chlorophyll $a$ in a few different solvents[42], and they found that the change of the 2DES spectra at short waiting times is dictated by the solvation dynamics. If the dipole moments of the ground and excited states differ from each other, the surrounding solvent molecules right after an instantaneous electronic excitation of a chromophore are in a non-equilibrium state in terms of solvation. Then, the solvent molecules would rearrange their orientations and move longitudinally from the excited state molecule to stabilize the newly created charge distributions of the solute molecule. Due to the ultrafast solvation dynamics, the SE peak that overlaps with the corresponding GSB peak at $T_w$ = 0 undergoes a frequency red-shift along the detection frequency axis[37], which makes the diagonal peak vertically broadened and elongated along the detection frequency axis. Therefore, the amplitude of the $GSB_0$ ($SE_0$) peak at (387 THz, 387 THz) ((387 THz, 375 THz)) shows a short-time decaying (rising) feature. Therefore, our observation that the short time changes of the $SE_0$ and $GSB_0$ signals are notably large suggests that the charge distribution of the $Q_{y,0}$ state differs from that of the ground state, $S_0$. This result is consistent with Moca et al.'s experimental observation[42]. To confirm this interpretation, we carried out density functional theory (DFT) and time-dependent DFT calculations of the ground ($S_0$) and excited ($Q_y$)

states of BChla, respectively. Indeed, these ab initio calculation results show that the dipole moments of these two states are distinctively different from each other (Supplementary Fig. 4). However, still, it is not entirely clear why the signals at SE$_1$ and GSB$_1$ generated by the transition to the $Q_{y,1}$ state induced by the pump field–matter interaction do not relax with $T_w$ for ~100 ps. One possible explanation is that the charge distribution of the $Q_{y,1}$ state is similar to that of the ground state, which makes the solvation dynamics contributions to the decays of the SE$_1$ and GSB$_1$ signals small.

**Femtosecond dynamics and vibrational coherences**. Using the short-time 2DES signals at four different points in a given 2DES spectrum, one can obtain the spectra of vibronically excited modes, which will be referred to as coherent vibrational spectrum (CVS) (Fig. 3e). We found four strongly vibronically coupled modes with frequencies of 190, 564, 728, and 1156 cm$^{-1}$. Here, the frequency resolution is estimated to be 8 cm$^{-1}$ by considering the scan range of the waiting time ($0 \leq T_w \leq 5.0$ ps) and the width (2.5 ps) of the apodization window. It should be emphasized that the SM-2DES technique does not need additional data acquisition time for a longer waiting time scanning to achieve an improved frequency resolution for the spectral analysis of vibrational quantum beats.

The vibrational mode at 728 cm$^{-1}$ is found to be the most strongly vibronically coupled to the $Q_y$ transition, which indicates that the displacement between the corresponding vibrational potential energy surfaces of the $S_0$ and $Q_y$ states is large. The vibronic coupling between the $Q_y$ transition and the 1156 cm$^{-1}$ mode is responsible for the $Q_{y,1}$ band because the frequency difference between the $Q_{y,0}$ and $Q_{y,1}$ bands in the absorption spectrum is close to 1156 cm$^{-1}$. The amplitudes of the other two modes at 190 and 564 cm$^{-1}$ depend on the excitation and detection frequencies. Although the information density of the three-dimensional 2DES data obtained with wide dynamic range 2DES measurements is very high, it is still difficult to thoroughly analyse the vibrational coherence spectra in Fig. 3e. Below, we shall present principal components analysis results and compare the DFT results for the ground and excited states of BChla.

**Nanosecond dynamics**. In Fig. 3f, we present the decay-constant distributions of the 2DES signals at the four peak positions, GSB$_0$, SE$_0$, GSB$_1$, and SE$_1$, which are the weighted histograms of the decay constants from the tri-exponential fitting analyses of the waiting-time-dependent signals. The amplitude of each exponential component is used as the weighting factor. From the total absorptive 2DES signals (Supplementary Fig. 5), we found that the lifetime of the $Q_y$ state is about 2.1 ns, which is consistent with the fluorescence lifetime of BChla in 1-propanol[43]. The sub-nanosecond components could be associated with the rotational diffusion of BChla because such a component was not observed in the magic-angle transient absorption study[44]. Although the nanosecond dynamics of the BChla solution is mainly determined by the rotational diffusion and population relaxation, the SM-2DES with nanosecond scanning capability could be useful for studying systems requiring both 2DES and nanosecond measurements in the future.

**Principal components analysis**. Vibrational coherence analysis at a single frequency point in the 2D map could cause a misinterpretation of the 2DES signal[45], even though the relative amplitudes and linewidths of vibronically coupled modes found in the CVS obtained by Fourier-transforming the 2DES signal with respect to $T_w$ provide crucial information on their vibrational coupling strengths and decoherences[46]. Therefore, it would be

useful to analyse both the coherent vibrational spectra and the 2D coherence maps to extract substantially detailed information on the nature and origin of coherent oscillations found in the spectrally congested $T_w$-resolved 2DES spectra. Here, we carried out a principal component analysis to identify the major 2D spectral patterns of the oscillating components in the 2DES signal and draw correlation between the vibronically coupled modes found in the CVS and the 2D spectra of major principal components. The entire three dimensional ($N_{\omega 1} \times N_{Tw} \times N_{\omega 2}$) 2DES data, where $N_{\omega 1}$, $N_{\omega 2}$, and $N_{Tw}$ are the numbers of data points of excitation frequency, detection frequency, and waiting time, respectively, are then represented with two 2D electronic spectra and two coherent vibrational spectra, which are the two principal components denoted as PC$_1$ and PC$_2$ in Fig. 4. All the other principal components are found to have negligibly small singular values.

Because the peak frequencies of the 2D coherence maps of the PC$_1$ and PC$_2$ overlap with the absorption and emission spectra of BChla, respectively, the majority of PC$_1$ can be identified with combination of GSB$_0$ and SE$_1$, and that of the PC$_2$ with SE$_0$ (see Supplementary Fig. 6 for detailed descriptions of these peak assignments and for the corresponding PC spectra extracted from the non-rephasing 2DES signals). There are six strong peaks at 190, 337, 564, 728, 901, and 1156 cm$^{-1}$ in the coherent vibrational spectra associated with the two principal components, PC$_1$ and PC$_2$, of the 2DES data (Fig. 4b). The relative amplitudes of the three peaks at 337, 564, and 901 cm$^{-1}$ in the CVS of the PC$_2$ are found to be much weaker than those in the CVS of the PC$_1$. Such detection-frequency dependences of the three peaks are notable when comparing the single-frequency point CVS at the maximum positions of the 2D coherence maps of the PC$_1$ and PC$_2$ (Fig. 4c). Consistent with the present PCA results on the vibronically coupled modes, we find that the amplitudes of the 2D coherence maps at 337, 564, and 910 cm$^{-1}$ (Supplementary Fig. 10) at (387 THz, 375 THz) are comparatively small. This indicates that the vibronic contributions from these modes to the SE$_0$ are negligible. The similarity and difference between the CVS of PC$_1$ and PC$_2$ suggest interesting information about the vibronic coupling strengths and their non-Condon effects, as will be discussed below.

**DFT calculation of vibronic coupling strength**. To assign the vibronically excited vibrational modes of BChla, we carried out two different DFT calculations to estimate the vibronic coupling strengths. The first approach, scheme **1**, is to calculate the coordinate displacements of all the normal modes using the normal mode projection method[46], which is based on direct comparisons of the optimized structures of the $S_0$ and $Q_y$ states (see the black bar spectrum in Fig. 4d). The second approach, scheme **2**, is based on the numerical calculation of the gradient of vibrational potential energy surface of the $Q_y$ state[46]. Once the slope of the vibrational potential energy curve at the equilibrium position of the ground state is obtained, one can estimate the vibrational coordinate displacement with invoking the harmonic approximation (red bar spectrum in Fig. 4d) (see Supplementary Note 6 for more details). Although these two approaches are not perfect, they still provide useful information about which modes are strongly coupled to the electronic $Q_y$ transition. From the DFT calculations, we could assign the six notable peaks in Fig. 4b at 190, 337, 564, 728, 901, and 1156 cm$^{-1}$ (see Supplementary Fig. 11 for their vibrational eigenvectors). The peak frequencies of the observed vibrational modes of BChla are consistent with the previous experimental results obtained with hole-burning spectroscopy[47] and 2DES[23–25] studies.

**Non-Condon effects and symmetry**. The electronic structures of the HOMO and LUMO are shown in Fig. 5a and b. The

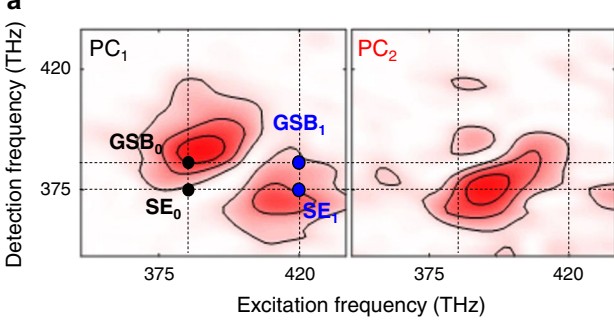

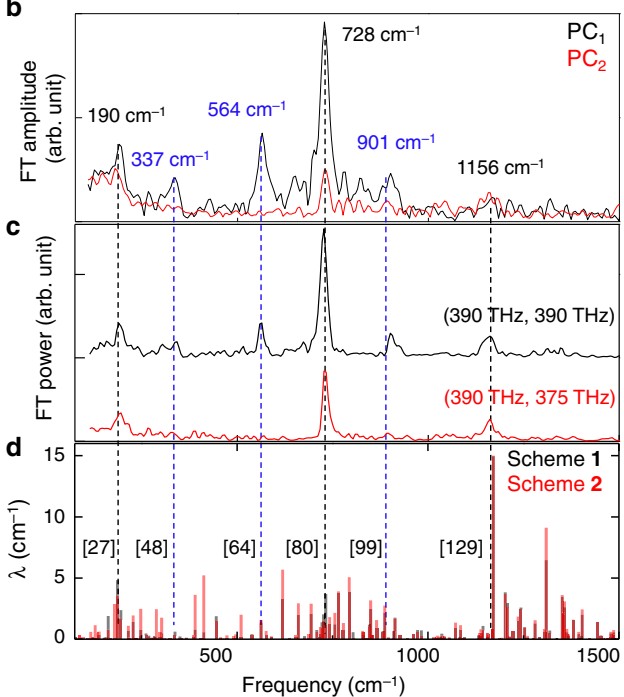

**Fig. 4 Principal components analysis of BChla 2DES. a** The 2D optical frequency spectra of two major principal components of the 2D coherent vibrational spectra for rephasing interaction pathway. **b** The vibrational spectra of **a** multiplied by their score. Relatively intense six peaks at 190, 337, 564, 728, 901, and 1156 cm$^{-1}$ are marked with their centre frequencies. **c** Coherent vibrational spectra obtained by Fourier-transforming the $T_w$-dependent 2DES signals at (390 THz, 390 THz) and (390 THz, 375 THz). The amplitudes of the three vibrational modes at 337, 564, and 901 cm$^{-1}$ are significantly small in the CVS from the 2DES signal close to the SE$_O$ position. **d** Theoretically calculated coherent vibrational spectra of BChla with respect to $Q_y$ excitation, where $\lambda$ indicates vibrational reorganization energy. The numbers in square brackets are the indices of the vibrational modes. See Supplementary Note 6 for more details on the simulation schemes **1** and **2**.

transition dipole moment (TDM) of the $Q_y$-transition is nearly parallel to the molecular y-axis ($\boldsymbol{\mu}_{eg}$ in Fig. 5a). From them, it becomes clear that the electronic transition from the $S_0$ to $Q_y$ states includes a $D_4$-to-$C_{2h}$ symmetry-breaking element. This symmetry analysis explains why the two modes with frequencies of 190 and 728 cm$^{-1}$ that have $D_{4h}$-to-$D_{2h}$ symmetry-breaking characters are vibronically coupled to the $S_0$-$Q_y$ electronic transition (Fig. 5c). However, the vibrational motions of the other four modes with frequencies of 337, 564, 901, and 1156 cm$^{-1}$ induce a $D_{4h}$-to-$C_s$ symmetry-breaking (Fig. 5d).

Recently, the non-Condon effects on the excitonic energy transfer processes in photosynthetic systems have been studied and discussed in the theoretical and 2DES studies[48–50]. In general, non-Condon effects refer to the dependence of the TDM on the vibrational coordinates. Such a breakdown of Condon approximation is already manifest in the broken mirror symmetry between the steady-state absorption and emission spectra in Fig. 3a[51]. However, the linear spectra do not provide information about which molecular vibrations have strong Herzberg-Teller vibronic couplings. For those non-Condon modes, the TDM at the equilibrium position ($Q'$) in the electronically excited state could differ from that at the equilibrium position ($Q^0$) in the ground state. In the 2DES experiments, such a non-Condon effect can be studied by examining the detection frequency-dependence of the CVS. As can be seen in Fig. 4b and c, the vibronic coupling strengths of the 337, 564, and 901 cm$^{-1}$ modes in the CVS associated with the SE$_0$ term are negligible, whereas they are strong in the CVS associated with the GSB$_0$ band. These disparities in the vibronic coupling strengths can be considered as experimental evidence of the non-Condon effects from these $D_{4h}$-to-$C_s$ symmetry-breaking modes. Here, it should be mentioned that the CVS of the GSB and SE contributions are similar to each other in the case of a dye (IR125) solution (Supplementary Fig. 7). Therefore, the difference between the two CVS in Fig. 4b and c is a characteristic feature of BChla. To understand how such a difference in the coherent vibrational spectra is related to the breakdown of the Condon approximation, let us consider the TDM associated with the electronic transition of BChla by probe beam interaction with BChla. The TDM determining the SE$_0$ signal should be evaluated at $Q'$, whereas that associated with the GSB$_0$ is the TDM at $Q^0$ (Fig. 5e). In the present work, we performed all-parallel polarization 2DES measurements. However, it would be useful to carry out polarization-controlled 2DES spectroscopic studies, e.g., 2DES anisotropy measurements[49], to extract more information about the relative directions of the absorptive and emissive TDMs ($\mu_{eg}(Q^0)$ and $\mu_{eg}(Q')$ in Fig. 5e) in the future. Our experimental results suggest that the vibronic TDMs are notably vibrational coordinate-dependent for the three (337, 564, and 901 cm$^{-1}$) modes, which cannot be revealed by linear spectroscopic studies of BChla. Noting that the molecular distortions along these three vibration coordinates ($Q_{sb}$) involve a $D_{4h}$-to-$C_s$ symmetry breaking, we anticipate that a further investigation about how such symmetry-breaking modes could affect the vibronic couplings and excitation transfer processes in photosynthetic reaction centres and light-harvesting complexes would be interesting.

## Discussion

In the present article, we showed that, as proof-of-principle experiments, direct time-domain measurement of the photon-echo response function of optical chromophores in solution is feasible using the SM-2DES technique. This measurement enables information on the correlation between two time-separated electronic coherences and femtosecond-resolved molecular relaxation occurring on femtosecond to nanosecond time scales to be obtained with a single-point radio frequency detector and without a metre-long mechanical delay line. This work is an experimental demonstration of SM-2DES for studying the excited-state dynamics of a photosynthetic pigment (BChla) in solution. The efficient 2DES measurement of BChla solution within wide time and frequency windows by SM-2DES enabled observation and analysis of characteristic excited-state vibrational coherences of BChla. The experimental results on BChla can be of use to understand the nature of the photosynthetic energy transfer processes. We anticipate that SM-2DES will be useful for investigating molecular structure changes of chemically reactive systems due to its femtosecond time-resolvability and wide dynamic range measurement capability. In addition, the use of

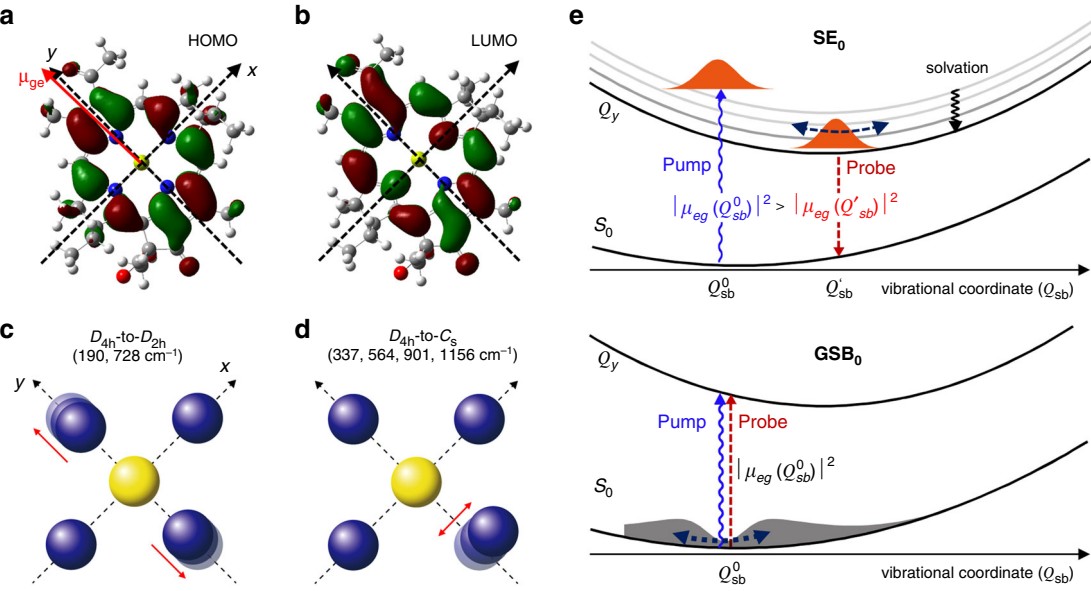

**Fig. 5 Symmetry-breaking modes and non-Condon effect.** HOMO (**a**) and LUMO (**b**) of BChla. The thick red arrow represents the $Q_y$ transition dipole moment ($\boldsymbol{\mu}_{eg}$). Blue and yellow balls represent nitrogen and magnesium atoms, while the grey balls do carbon atoms. x- and y-axes are defined from two orthogonal N–Mg–N lines. Thin red arrows emphasizes the nuclear motions of N atoms in the BChla centre. The key eigenvector elements of the $D_{4h}$-to-$D_{2h}$ (**c**) and $D_{4h}$-to-$C_s$ (**d**) symmetry-breaking vibrational modes at the BChla centre are shown here. The full vibrational eigenvectors of these six modes are presented in Supplementary Fig. 11. The red arrows in **c** and **d** show the directions (eigenvector elements) of nitrogen atoms. **e** Schematic diagram describing a non-Condon effect on the reduced vibrational amplitudes of the three symmetry-breaking modes found in the CVS from the 2DES signal at (390 THz, 375 THz). Due to the destructive interference between the Franck–Condon factor and the Herzberg–Teller vibronic coupling, the oscillator strength associated with the $Q_{y,0}$-$S_0$ transition at the equilibrium coordinate $Q'_{sb}$ in the excited state is smaller than that at the equilibrium coordinate $Q^0_{sb}$ in the ground state. Such a non-Condon effect is also manifest in the broken mirror symmetry between the steady-state absorption and emission spectra in Fig. 3a.

multiple single-point detectors would make polarization-dependent 2DES measurements efficient, which in turn could provide information on field-induced optical anisotropy of rotating molecules and electronic or vibrational optical activity of chiral molecules[52] in condensed phases. We believe that a few immediate applications of SM-2DES are not only to study excitonic dynamics in the manifold of one and multiple exciton states in solar harvesting complexes, semiconductors, and quantum dots but also to characterize the 2D optical response of chromophores in biological systems.

## Methods

**Synchronized mode-locked laser-based 2DES.** Two Ti:Sapphire lasers (Rainbow 2, Femtolasers) producing a train of pulses with a duration time of 7 fs were used for the SM-2DES setup. The repetition frequencies of the MLs were phase-locked to a reference radio frequency (RF) signal from an 8-channel frequency synthesizer (Holzworth, HS9008A), the time-base of which is linked phase-coherently to a GPS-disciplined Rb atomic clock. The repetition frequencies of $ML_1$ and $ML_2$ were adjusted to be 80 MHz + $\Delta f_r$ and 80 MHz, respectively. The waiting time $T_w$ was automatically scanned due to the difference in the repetition frequencies, $\Delta f_r$, between the two MLs, which is the key aspect of the ASOPS technique. The sampling rate was synchronized to the repetition rate of $ML_2$ (80 MHz) to record every pulse-to-pulse intensity difference of the signal. We set $\Delta f_r$ to be either 38.4 Hz or 3.2 kHz, which corresponds to $\Delta T_w$ values of 6 fs and 500 fs, respectively, for the SM-2DES measurements. Fractions of $ML_1$ and $ML_2$ are non-colinearly focused at a nonlinear crystal to generate a sum-frequency-generation field, which is utilized as an optical trigger for the automatic $T_w$-scan. The time jitter of the $T_w$-scan is determined by the pulse shape of the optical trigger and it was <1 fs. Due to the fast automatic scanning of $T_w$, $\tau_2$ can be scanned continuously by using a mechanical delay stage. The time interval for the $\tau_1$ step-scan was controlled to be $\Delta\tau_1 = 1.868$ fs, which corresponds to a translational stage step of 0.14 μm for each continuous $\tau_2$-scan. The time resolution of our SM-2DES was 12.5 fs, which was estimated by measuring the cross-correlation between $ML_1$ and $ML_2$ at the sample position (see Supplementary Note 1 for details).

IR125 and BChla were purchased from Exciton and ChemCruz, respectively. The two molecules were used without further purification. IR125 (BChla) was dissolved in ethanol (1-propanol), and the light absorbance of the solution was adjusted to be ~0.2 (0.3), where the thickness of the sample cell is 200 μm. The thickness of the front window of the sample cell was as thin as 100 μm, which was

to minimize additional pulse broadening by the optical cell itself. The solution sample continuously flowed with a gear pump (Micropump) during the SM-2DES measurements. The gear pump was cooled down to 20 °C with a circulating chiller to protect the sample from heat generated from the gear pump. The ASOPS-based time-resolved spectroscopy inevitably wastes a lot of pulses because the automatic time scan involves significantly longer delays than electronic and vibrational relaxation times. Therefore, the duty cycle of such an ASOPS-based time-resolved vibrational spectroscopy of chromophores in the condensed phases could be pretty low. For example, if the time span is 5 ps for measuring vibrational dephasing constants and the pulse repetition period is 12.5 ns ($f_r = 80$ MHz) the duty cycle[29] is estimated as 5 ps/12.5 ns, which is 0.04%. In this case, to protect the sample from photochemical and thermal damages caused by the interactions of the remaining 99.96% of pulses from each ML with the sample, it is necessary to block them by using a laser shutter synchronized with optical triggers. In contrast, a much longer scanning time (3.25 ns in this study) is required for the fast-scan-mode SM-2DES with $\Delta f_r = 3.2$ kHz ($\Delta T_w = 0.5$ ps) to characterize slow electronic relaxation dynamics. In this case, the duty cycle is as high as 26.0%. In order to minimize the optical damage on the sample, the incident beams were blocked by a set of optical shutters when data recording is not necessary. As a result of this gated sampling, the degradation rate of Bchla became three times slower (see Supplementary Note 5 for more details). We checked that there was no additional signal from the sample degradation.

**Phase correction and amplitude calibration.** Small portions of the local oscillator fields from $ML_2$ before and after the $\tau_2$-scanning stage were taken using two beam splitters and were used to construct a Mach–Zehnder interferometer. This process provides the reference interferograms that are synchronized with the SM-2DES data collections. The reference interferograms are used to compensate for the phase fluctuation error in the detection frequency and the nonorthogonality between $T_w$- and $\tau_2$-scans.

In interferometric SM-PP spectroscopy, the probe field itself provides a phase reference automatically, and the spectral amplitudes of the measured PP spectrum can be simply normalized by using the pulse spectrum[34]. On the other hand, SM-2DES signals cannot be normalized, and their electric field phases cannot be easily corrected. However, the phase and amplitude of SM-2DES can be analytically calibrated and measured, respectively, by using interferometrically measured SM-PP spectra with exactly the same instrument (see Supplementary Note 3). The calibration factor is the ratio of the integrated SM-2DES spectrum over the excitation frequency to the interferometrically measured SM-PP spectrum. Unlike the conventional approach using the projection theorem, the present calibration and phase correction scheme

works well because the complex (both real and imaginary) SM-PP signals can be obtained and used as both amplitude and phase references.

## Data availability

The data that support the findings of this study are available within the paper and its supplementary information files. Raw data are available from the corresponding author on a reasonable request.

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

## Acknowledgements

This work was supported by the Institute for Basic Science (IBS-R023-D1).

## Author contributions

M.C., T.H.Y., and JW.K. conceived this experiment. JW.K. performed the experiments and J.J. developed a theory of SM-2DES. JW.K. performed the coherent vibrational spectroscopy simulation of BChla. JW.K., T.H.Y., and M.C. interpreted the experimental results. All the authors contributed to the writing of the manuscript.

## Competing interests

The authors declare no competing interests.
