## [Peer Review File · Nature Communications]

Editorial Note: This manuscript has been previously reviewed at another journal that is not operating a transparent peer review scheme. This document only contains reviewer comments and rebuttal letters for versions considered at Nature Communications. Mentions of the other journal and prior referee reports have been redacted.

REVIEWERS' COMMENTS

Reviewer #1 (Remarks to the Author):

I had already reviewed manuscript NCOMMS-20-27941-T when it was submitted to [redacted]. I have read in details the revised manuscript NCOMMS-20-27941-T [redacted]. I was already in favor of the publication of the manuscript in [redacted]. I can only give my strongest recommendation for it to be accepted as is to Nature Communications. To me, the manuscript compares very favorably with the papers published in Nature Communications that are within my field of expertise.

In ms. NCOMMS-20-27941-T, the experimental concept is novel and compelling. The experiment appears to be cleverly and rigorously performed. The experimental demonstration is highly convincing and presented honestly, in a balanced way. The discussion is inspiring and clearly identifies new opportunities that can be investigated with the new technique of SM-2DES. Through the multiple rounds of reviews, the authors have come up with a manuscript that is incredibly well written and well presented. It is both accessible to non-specialists and detailed and insightful to answer the interrogations of other specialists in the field and to allow duplication of the set-up.

I also believe that it is of prime importance for spectroscopists working in chemistry and in physics that new instrumental techniques are proposed and thoroughly characterized. This is what the authors do in this manuscript with their instrument of multidimensional spectroscopy to study photon-echoes. Proof-of-principle experiments are necessary to evaluate and quantify the potential of a new experimental technique and to assess its novelty. Progress in the understanding of the structure and dynamic of matter relies on the ingenuity of scientists to invent instruments that measure what could not be measured before, or that significantly improve existing capabilities. From this point of view, instrumentation for spectroscopy is of broad and multidisciplinary interest. It is key to ground-breaking studies in chemistry.

I am deeply convinced that NCOMMS-20-27941-T establishes a new technique that will inspire many scientists working in chemistry and in physics. As a scientist working in the field of frequency-comb spectroscopy for many years, I also see NCOMMS-20-27941-T as an important milestone for the spread of frequency-comb spectroscopy to chemistry and biology, where the huge potential is still largely uncovered. This is excellent work and I am sure NCOMMS-20-27941-T will be a high-impact paper.

As I said above, [redacted], with the strongest enthusiasm, I have no hesitation in recommending its acceptance by Nature Communications.

Reviewer #2 (Remarks to the Author):

The authors have yet again substantially improved their manuscript and clarified some issues. However, I still have some reservations regarding the appearance and analysis of the data, as I do not think that presented data and analysis provide substantial support for the conclusions written in the manuscript. That being said, I find the developed method very elegant, which most likely will find applications and will help to achieve deep understanding of photophysical processes. I would also like to remark that it is rare to come across another paper, which would match the tremendous effort invested in the present project and manuscript. In summary, as a technical demonstration of the method, which is expected to be of interest to a broad audience, I can recommend it to be published in Nature Communications.

Some minor remarks

n , m , and p , which are the frequency comb mode numbers, should be defined in the text.

Line 121. Expression "relaxation of the transient grating in the direction..." is not accurate and is perhaps

lacking something.

Line 224, in general, laser spectrum effects 2D spectra on both excitation and detection frequencies, and not only on the detection frequency as stated.

Lines 284-286 The provided possible explanation for the lack of the 100 ps relaxation for the SE1 signal is inaccurate. SE1 signal originates from the $Q_{y,0}$ state and not from $Q_{y,1}$ state as written.

The sentence starting with "Fractions of..." on the line 459 is not complete.

Lines 496-498. I do not think that the statement "Thus, conventional 2DES techniques utilizing multiple pulses and translational stages cannot provide absolutely quantitative 2DES data even though the phase of the signal electric field could be numerically corrected by matching the projected 2DES spectra to the independently measured PP spectra" is absolutely correct. It should be possible to do so, as I believe all the necessary information for obtaining absolutely quantitative 2DES data is in principle available, especially from the pump-probe based 2D setups. This requires measuring PP with the same instrument under the same conditions, just like in the presented work.

Authors' point-by-point responses to reviewers' comments and suggestions

Reviewer #1 (Remarks to the Author):

I had already reviewed manuscript NCOMMS-20-27941-T when it was submitted to [redacted] . I have read in details the revised manuscript NCOMMS-20-27941-T [redacted]

I was already in favor of the publication of the manuscript in [redacted]. I can only give my strongest recommendation for it to be accepted as is to Nature Communications.

To me, the manuscript compares very favorably with the papers published in Nature Communications that are within my field of expertise.

In ms. NCOMMS-20-27941-T, the experimental concept is novel and compelling. The experiment appears to be cleverly and rigorously performed. The experimental demonstration is highly convincing and presented honestly, in a balanced way. The discussion is inspiring and clearly identifies new opportunities that can be investigated with the new technique of SM-2DES. Through the multiple rounds of reviews, the authors have come up with a manuscript that is incredibly well written and well presented. It is both accessible to non-specialists and detailed and insightful to answer the interrogations of other specialists in the field and to allow duplication of the set-up.

I also believe that it is of prime importance for spectroscopists working in chemistry and in physics that new instrumental techniques are proposed and thoroughly characterized. This is what the authors do in this manuscript with their instrument of multidimensional spectroscopy to study photon-echoes. Proof-of-principle experiments are necessary to evaluate and quantify the potential of a new experimental technique and to assess its novelty. Progress in the understanding of the structure and dynamic of matter relies on the ingenuity of scientists to invent instruments that measure what could not be measured before, or that significantly improve existing capabilities. From this point of view, instrumentation for spectroscopy is of broad and multidisciplinary interest. It is key to ground-breaking studies in chemistry.

I am deeply convinced that NCOMMS-20-27941-T establishes a new technique that will inspire many scientists working in chemistry and in physics. As a scientist working in the field of frequency-comb spectroscopy for many years, I also see NCOMMS-20-27941-T as an important milestone for the spread of frequency-comb spectroscopy to chemistry and

biology, where the huge potential is still largely uncovered. This is excellent work and I am sure NCOMMS-20-27941-T will be a high-impact paper.

As I said above,[redacted], with the strongest enthusiasm, I have no hesitation in recommending its acceptance by Nature Communications.

Authors' Reply. We are really grateful to the reviewer #1 for her/his strong support for our work and recommendation of our manuscript for publication in Nature Communications.

Reviewer #2

The authors have yet again substantially improved their manuscript and clarified some issues. However, I still have some reservations regarding the appearance and analysis of the data, as I do not think that presented data and analysis provide substantial support for the conclusions written in the manuscript. That being said, I find the developed method very elegant, which most likely will find applications and will help to achieve deep understanding of photophysical processes. I would also like to remark that it is rare to come across another paper, which would match the tremendous effort invested in the present project and manuscript. In summary, as a technical demonstration of the method, which is expected to be of interest to a broad audience, I can recommend it to be published in Nature Communications.

Reply. We thank the reviewer #2 for recommending our manuscript to be published in Nature Communications.

Comment 1. n , m , and p , which are the frequency comb mode numbers, should be defined in the text.

Reply 1. We have added a sentence defining the indices in page 7 as

“Here, n , m , and p are the frequency comb mode numbers.”

Comment 2. Line 121. Expression “relaxation of the transient grating in the direction...” is not accurate and is perhaps lacking something.

Reply 2. We have corrected the sentence as

“... relaxation of the transient grating diffracted in the direction ...”.

Comment 3. Line 224, in general, laser spectrum affects 2D spectra on both excitation and detection frequencies, and not only on the detection frequency as stated.

Reply 3. We believe that the reviewer #2’s comment is on the sentence in line 244 not 224 in the previous manuscript). The reviewer is correct on this point. In general, both the pump (excitation) and probe (detection) laser spectra would influence the two-dimensional spectral features of each 2DES spectrum. Therefore, we have added one more sentence mentioning

this complication to the revised manuscript

“Note that, in general, both the pump and probe spectra affect the spectral features of the 2DES spectrum along the excitation and detection frequencies.”

Comment 4. Lines 284-286 The provided possible explanation for the lack of the 100 ps relaxation for the SE_1 signal is inaccurate. SE_1 signal originates from the $Q_{y,0}$ state and not from $Q_{y,1}$ state as written.

Reply 4. We agree with the reviewer on this point. The corresponding sentence is rewritten as

“...it is not entirely clear why the signals at SE_1 and GSB_1 generated by the transition to the $Q_{y,1}$ state do not relax with T_w for approximately 100 ps.”.

Comment 5. The sentence starting with “Fractions of...” on the line 459 is not complete.

Reply 5. We thank the reviewer #2. The corresponding sentence is rewritten as

“Fractions of ML_1 and ML_2 are non-collinearly focused at a nonlinear crystal to generate a sum-frequency-generation field, which is utilized as an optical trigger for the automatic T_w -scan.”

Comment 6. Lines 496-498. I do not think that the statement “Thus, conventional 2DES techniques utilizing multiple pulses and translational stages cannot provide absolutely quantitative 2DES data even though the phase of the signal electric field could be numerically corrected by matching the projected 2DES spectra to the independently measured PP spectra” is absolutely correct. It should be possible to do so, as I believe all the necessary information for obtaining absolutely quantitative 2DES data is in principle available, especially from the pump-probe based 2D setups. This requires measuring PP with the same instrument under the same conditions, just like in the presented work.

Reply 6. We agree with the reviewer on this particular point. Therefore, we have removed the corresponding sentence from the manuscript. We thank the reviewer #2 for this clarification. Again, we thank her/him for recommending our manuscript to be published in Nature Communications.